# Interpretable Boundary-based Watermark Up to the condition of Lovász Local Lemma

## Abstract

Watermarking techniques have emerged as pivotal safeguards to defend the intellectual property of deep neural networks against model extraction attacks. Most existing watermarking methods rely on the identification of samples within randomly selected trigger sets. However, this paradigm is inevitably disrupted by the ambiguous points that exhibit poor discriminability, thus leading to the misidentification between benign and stolen models. To tackle this issue, in this paper, we propose a boundary-based watermarking method that enhances the discernibility of trigger set, further improving the ability in distinguish benign and stolen models. Specifically, we select trigger samples on the decision boundary of base model and assigned them labels with the least probabilities, while providing a tight bound based on the Lovász Local Lemma. This approach ensures the watermark's reliability in identifying stolen models by improving discriminability of trigger samples. Meanwhile, we provide theoretical proof to demonstrate that the watermark can be effectively guaranteed under the constraints guided by the Lovász Local Lemma. Experimental results demonstrate that our method outperforms the state-of-the-art watermarking methods on CIFAR-10, CIFAR-100 and ImageNet datasets. Code and data will be released publicly upon the paper acceptance.

## 1 Introduction

Deep learning methods have demonstrated remarkable success in innumerable industrial contexts, such as computer vision (He et al., 2016; Dosovitskiy et al., 2020) and natural language processing (Brown et al., 2020; Vaswani et al., 2017). However, deep neural networks (DNNs) entail substantial costs, primarily due to two critical factors: the heavy reliance on extensive training data (Halevy et al., 2009) and substantial computational resources (Floridi & Chiriatti, 2020). Training and deploying high-performing DNNs is both time-consuming and resource-intensive. Consequently, it is crucial to protect the intellectual property of model providers, especially when users are permitted to query and access the outputs of these valuable models deployed on the cloud platforms.

While model providers can protect their intellectual property by keeping model parameters confidential, they are obligated to provide APIs for user access as service providers. This exposes the model to potential threats from adversaries who can extract the functionality of DNN models using black-box model extraction attacks (Tramèr et al., 2016; Orekondy et al., 2019). In this scenario, adversaries engage in extensive querying of the victim model to acquire labels for a surrogate dataset, which is then utilized to train their replica model, effortlessly creating a stolen copy of the original model (Pal et al., 2019; Papernot et al., 2017). Since adversaries' inputs are common and similar to those of benign users, model providers struggle to differentiate between them. As a result, all outputs are exposed to adversaries, making it challenging to prevent model extraction attacks solely through the API interaction process.

To tackle model extraction attacks, researchers have introduced model watermarking techniques (Kahng et al., 1998). Early works followed the parameter-embedding watermarking paradigm (Uchida et al., 2017; Darvish Rouhani et al., 2019; Kuribayashi et al., 2021; Mehta et al., 2022), where the watermark is embedded into the model parameters. However, these approaches are limited to white-box scenarios, and it is impossible to verify the watermark if suspicious models disclose their parameters. To this end, recent studies have shifted their focus to backdoor-based

watermarking techniques (Adi et al., 2018; Zhang et al., 2018; Li et al., 2019; Namba & Sakuma, 2019; Zhang et al., 2020a;b; Chen et al., 2021; Yang et al., 2021; Jia et al., 2021; Maini et al., 2021; Li et al., 2022; Bansal et al., 2022; Kim et al., 2023; Yu et al., 2023; Li et al., 2024; Pautov et al., 2024) and have achieved promising results in black-box scenarios. In backdoor-based methods, defenders select specific input-output pairs $(\tilde{x}, \tilde{y})$ as the trigger set and train the model to overfit to this trigger set. To claim ownership of the model, defenders query the suspicious model with these specific inputs $\tilde{x}$ and verify whether the returned results match the predefined set $\tilde{y}$. By using different backdoor algorithms, these studies successfully improve the ability of stolen models to recognize trigger samples. Moreover, several certifiable watermarking techniques (Bansal et al., 2022; Jiang et al., 2023; 2024) provide theoretical guarantees. These methods perturb the model parameters to increase robustness, and the watermark is guaranteed to be effective unless adversaries change the model parameters beyond a certain $\ell_2$ distance.

However, previous works predominantly rely on "random" configurations for trigger sets: they either randomly sample instances (Kim et al., 2023), assign random labels to triggers (Zhang et al., 2018; Yu et al., 2023), or combine both strategies (Zhang et al., 2018; Jia et al., 2021; Bansal et al., 2022; Li et al., 2022). This approach may lead to the selection of low-quality points and makes it difficult to accurately distinguish between benign models and stolen models. Specifically, randomly chosen samples that cluster around central points, a common occurrence in dense clusters, can adversely impact neighboring samples and consequently deteriorate the model's performance. When the randomly selected label aligns with the output of the benign models, it presents a hurdle in identifying a stolen model. Additionally, in terms of theoretical guarantees, existing random smoothing-based methods (Bansal et al., 2022; Jiang et al., 2023; 2024) impose constraints on all model parameters, introducing calculations involving a multitude of extraneous parameters, consequently diminishing the theoretical boundaries that can be assured. Furthermore, these assurances are tailored to random smoothing techniques and do not offer a universal guarantee for alternative methodologies.

To address the above issues, we propose a boundary-based watermarking method that meticulously selects boundary samples and their corresponding labels. Specifically, we dissect the identification of whether a suspicious model is benign or stolen into two pivotal components: the discriminability of the trigger set and the probability of $(\tilde{x}, \tilde{y})$ in distinguishing between benign and stolen models. Correspondingly, we devise a trigger set selection strategy based on this analysis to enhance both components simultaneously. This strategy comprises two processes: Initially, we train a base model and select samples near the decision boundary as trigger set. Subsequently, we perturb the base model to simulate multiple independently trained models, then query these perturbed models and select the least frequently occurring label for each trigger sample. This process ensures that the resulting trigger set can tolerantly identify benign and stolen models. Furthermore, we provide a tighter bound by constraining watermark-related parameters guided by the Lovász Local Lemma. Under such constraints, the effectiveness of the watermark can be guaranteed by incorporating multiple watermarking pairs.

Our contributions are summarized as follows:

- We propose a boundary-based watermarking method that defends against model extraction attacks by meticulously selecting boundary samples as the trigger set and assigning permissive labels.

- We provide a tighter theoretical bound under the guidance of the Lovász Local Lemma. The watermark can be guaranteed by constraining the related parameters.

- We conduct extensive experimental evaluations on three public datasets to validate the effectiveness of the proposed method, obtaining state-of-the-art performance on both trigger set accuracy and p-value.

## 2 RELATED WORK

### 2.1 WATERMARKS

Watermarking techniques aim to protect intellectual property against the theft of elaborate models (Kahng et al., 1998). Traditional deep learning based methods embedded the watermarks into the parameters (Uchida et al., 2017; Chen et al., 2019; Darvish Rouhani et al., 2019), necessitating

parameter access for watermark verification. However, given that the owners of suspicious models may not grant access to the parameters, hindering the use of white-box methods for ownership identification, black-box scenarios emerge as a more desirable approach.

Due to the remarkable performance in ownership verification of stolen models, recent works focusing on backdoor-based watermarking have drawn much attention. These methods operate by training the model to fit outlier input-output pairs exclusively known to the defender, which can subsequently be utilized to assert ownership of the model. The backdoor-based work can be roughly divided into two categories. One paradigm involves selecting random trigger sets and introducing novel loss functions to enhance the robustness of classification on trigger samples, such as expanding the trigger's neighborhood (Kim et al., 2023) and entangling normal data with trigger data (Jia et al., 2021). The other approaches focus on the selection of meticulously chosen triggers, primarily derived from out-of-distribution (OOD) data (Zhang et al., 2018; Li et al., 2022; Yu et al., 2023). Furthermore, MAT (Li et al., 2024) selects trigger sets from the decision boundary and assigns the second probable class as classification labels. However, this selection approach exhibits a low tolerance for classification errors. Specifically, under various training settings (e.g. random initialization or different optimization methods), the models can easily misclassify these samples as the category that should originally have the second highest confidence, thus may be erroneously classified as stolen models. Conversely, our method improves distinguishability of trigger set by selecting samples from the decision boundary and assigning labels that rarely appear in the predictions of benign models. When a suspicious model exposes a small number of watermarked input-output pairs, the defender can confidently assert ownership of this model.

In terms of theoretical proofs, various watermarking methods have been proposed to prove their certificates, with the majority relying on random smoothing technique (Goldberger et al., 2020; Bansal et al., 2022; Jiang et al., 2023; 2024). Nevertheless, these methods necessitate the adherence of the stolen model's parameters to a fixed $\ell_2$ threshold, which can be easily circumvented by adversaries through diverse tricks such as selecting random seeds, adapting optimizers, or even fine-tuning hyperparameters. In this paper, we alleviate this restriction by adapting Lovász Local Lemma into the selection of trigger sets. And our watermark is guaranteed to be unremovable when the watermark-related neurons satisfy the conditions of the Lovász Local Lemma and the size of the trigger set is not small.

## 2.2 MODEL EXTRACTION ATTACK

Model extraction attack (Tramèr et al., 2016), as a functionality stealing method (Orekondy et al., 2019), is currently considered the most powerful black-box attack (Lukas et al., 2022). In these attacks, adversaries typically start by collecting or synthesizing an initially unlabeled surrogate dataset. For example, Papernot et al. (2017) utilized Jacobian-based dataset augmentation to increase the potency of attacks. Tramèr et al. (2016) proposed three techniques for uniform data sampling. They assign labels for these samples by querying the original model and then use this surrogate dataset to train a stolen copy of the victim model.

Watermarking techniques designed to combat such attacks intertwine the training of both the dataset and the trigger set, facilitating the surrogate model in comprehending the watermarked decision boundary. However, the presence of watermarked decision boundaries may impact the model's performance, leading to a trade-off in all approaches between the model's practical utility for legitimate users and the success of watermark transfer (Alabdulmohsin et al., 2014; Shukla, 2020; Lee et al., 2019; Tramèr et al., 2016). Our approach mitigates this dilemma while ensuring the effective watermarking by meticulously selecting boundary samples and assigning them permissive labels.

## 3 METHOD

In this section, we elaborate on our principal concepts for selecting trigger samples and assigning corresponding labels, followed by presenting our watermarking algorithm and theoretical analysis.

## 3.1 PRELIMINARIES

### 3.1.1 MODEL EXTRACTION AND TRIGGER SET-BASED WATERMARKING

We begin by introducing trigger set-based watermarking methods, which form the foundation of our approach. In our work, we consider classification problem with $K$ classes. Given a training dataset $\mathcal{D} = \{(x_i, y_i)\}_{i=1}^n$ where $x_i \in \mathbb{R}^d$ and $y_i \in \{1, \cdots, K\}$, we train a base model $M_b$ to minimize the empirical risk as:

$$\mathcal{L}(\mathcal{D}) = \frac{1}{|\mathcal{D}|} \sum_{(x,y) \in \mathcal{D}} \ell\left(M(x) - y\right)$$

where $\ell$ is a loss function such as cross-entropy.

If the model lacks protective measures, an adversary can attempt to steal its functionality without accessing the dataset $\mathcal{D}$. Specifically, the adversary first queries the source model with a sample $\hat{x}_i$, obtains source model's output $M(\hat{x}_i)$, and then trains a surrogate model $\hat{M}$ to replicate the functionality of the source model using the surrogate dataset $\hat{\mathcal{D}} = \{(\hat{x}_i, \hat{y}_i)\}_{i=1}^m$ by minimizing the loss function

$$\min \hat{\mathcal{L}}(\hat{\mathcal{D}}) = \min \frac{1}{|\hat{\mathcal{D}}|} \sum_{(x,y) \in \hat{\mathcal{D}}} \hat{\ell}\left(M(x) - \hat{M}(x)\right) \tag{1}$$

where $\hat{\ell}$ often uses the Kullback-Leibler divergence (Thomas & Joy, 2006). This model extraction attack, as a special case of knowledge distillation (Hinton, 2015), allows the adversary to steal the functionality of the model trained on $\mathcal{D}$ without direct access to $\mathcal{D}$ itself (Orekondy et al., 2019).

To protect intellectual property, the owner of the source model can embed a watermark into the model and verify whether a suspicious model contains the same watermark. One method of watermarking is to use a trigger set, which also forms the foundation of our approach. Specifically, the owner of the source model randomly samples $\{(x_i, y_i)\}_{i=1}^m$ from $\mathcal{D}$ and replaces the original labels $y_i$ with new labels $\tilde{y}_i \neq y_i$, producing the trigger set $\tilde{\mathcal{D}} = \{(x_i, \tilde{y}_i)\}$. The training data is changed from $\mathcal{D}$ to a combination of clean data $\mathcal{D}_c = \mathcal{D} \backslash \{(x_i, y_i)\}_{i=1}^m$ and the trigger set $\tilde{\mathcal{D}}$. The model owner trains the watermarked model by minimizing the loss over both datasets:

$$\min \mathcal{L}_c(\mathcal{D}_c) + \tilde{\mathcal{L}}(\tilde{\mathcal{D}})$$

where $\mathcal{L}_c$ and $\tilde{\mathcal{L}}$ typically use different loss functions to fit the trigger set as closely as possible, ensuring that the watermark is not removed as an outlier during model extraction attacks.

### 3.1.2 THE LOVÁSZ LOCAL LEMMA

The Lovász Local Lemma (Erdos & Lovász, 1975) is a powerful tool to non-constructively prove the existence of combinatorial objects meeting a prescribed collection of criteria. Let $\mathcal{A}$ be a finite collection of mutually independent events in a probability space. The probability that none of these events happen is exactly $\prod_{A_i \in \mathcal{A}}(1 - \Pr[A_i])$. In particular, this probability is positive whenever no event in $\mathcal{A}$ has probability 1. László Lovász's famous Local Lemma (Erdos & Lovász, 1975) relaxes the independence condition slightly: as long as the events are "mostly" independent, but still concludes that with positive probability none of the events happen if the individual events have bounded probability. Here is the lemma in a general form.

**Theorem 1** (Erdos & Lovász (1975)). *Let $\mathcal{A}$ be a finite set of events in a probability space. For $A_i \in \mathcal{A}$ let $\Gamma(A_i)$ be a subset of $\mathcal{A}$ satisfying that $A_i$ is independent from the collection of events $\mathcal{A} \backslash (A_i \cup \Gamma(A_i))$. If there exists an assignment of reals $\alpha : \mathcal{A} \to (0, 1)$ such that*

$$\Pr[A_i] \leq \alpha(A_i) \prod_{A_j \in \Gamma(A_i)} (1 - \alpha(A_j)), \quad \forall A_i \in \mathcal{A}, \tag{2}$$

*then the probability of avoiding all events in $\mathcal{A}$ is positive, in particular*

$$\Pr\left[\overline{A_1} \wedge \cdots \wedge \overline{A_n}\right] \geq \prod_{i \in \{1, \cdots, n\}} (1 - \alpha(A_i)). \tag{3}$$

Figure 1: Illustration of the overall architecture of the proposed method. Given a base model, our watermarking method selects trigger samples and assigns them labels according to a newly designed selection strategy. Then the stolen model can be identified based on the predictions of the selected trigger samples.

Our method proves a tight bound by constraining watermark-related parameters, guided by the Lovász Local Lemma. Unlike random smoothing-based approaches (Bansal et al., 2022), which preserve the watermark by limiting parameter modifications within a given $\ell_2$ distance across the entire network, our approach constrains only the parameters activated by the watermark (Jia et al., 2021). When these constraints are satisfied, the watermark can be embedded with a positive probability and differentiate between benign and stolen models, unless the size of trigger set is too small (Corollary 2.1).

## 3.2 OUR WATERMARKING METHOD

Before presenting our method, we first decompose the overall probability of successfully determining whether a model $M$ is benign or stolen into two components: ( 1) the accuracy of classifying the trigger samples (Trigger set acc. in Table 1), and (2) the probability that $(\tilde{x}, \tilde{y})$ can identify benign or stolen models. The overall probability can be expressed as the product of these two factors:

$$\Pr[M \text{ is stolen model}] = \frac{1}{|\tilde{D}|} \sum_{(\tilde{x},\tilde{y}) \in \tilde{D}} \mathbb{1}[M(\tilde{x}) = \tilde{y}] \cdot \Pr[\text{diff}(\tilde{x}, \tilde{y})], \tag{4}$$

where $\mathbb{1}$ is the indicator function, $\mathbb{1}[M(\tilde{x}) = \tilde{y}]$ indicates whether the model $M$ correctly predicts the label $\tilde{y}$ for the trigger $\tilde{x}$ (Note that the trigger set accuracy is typically calculated as $\frac{1}{|\tilde{D}|} \sum_{(\tilde{x},\tilde{y}) \in \tilde{D}} \mathbb{1}[M(\tilde{x}) = \tilde{y}]$). $\text{diff}(\tilde{x}, \tilde{y})$ represents the event that $(\tilde{x}, \tilde{y})$ successfully differentiates between a benign model and a stolen one, and can be approximated by the probability where $M(\tilde{x}) \neq \tilde{y}$ within the predictions of the benign model $M$.

Here, we provide two examples to clarify why we split this probability into two components. In the first instance, consider a trigger $(\tilde{x}_1, \tilde{y}_1)$. When querying 10,000 benign models for the same task with $\tilde{x}$, 5,000 models yield $\tilde{y}_1$, while the remaining models output different results $y' \neq \tilde{y}_1$. Such scenarios frequently occur near decision boundaries, resulting in $\Pr[\text{diff}(\tilde{x}_1, \tilde{y}_1)] \approx 0.5$. In this scenario, even though the accuracy of the trigger set improves when the suspicious model correctly identifies the trigger, a probability of only 0.5 is insufficient to confidently indicate that the model is stolen. In the second instance, consider a trigger $(\tilde{x}_2, \tilde{y}_2)$. When querying 10,000 benign models, none of them output $\tilde{y}_2$, indicating that $(\tilde{x}_2, \tilde{y}_2)$ can effectively identify the benign and stolen models, resulting in $\Pr[\text{diff}(\tilde{x}_2, \tilde{y}_2)] \approx 1$. Subsequently, if both our watermarked model and the suspicious model produce $\tilde{y}_2$, it strongly suggests that the suspicious model is stolen only based on $(\tilde{x}_2, \tilde{y}_2)$.

As illustrated in Figure 1, our method considers both components, improving trigger set accuracy and enhancing the ability to identify benign and stolen models. First we focus on the selection

---

**Algorithm 1:** Procedure of Generating Trigger Set

---

**Input** : Base model $M_b$, number of perturbed models $s$, perturbation range $\delta$, boundary thresholds $a$ and $b$, size of trigger set $m$;

**Output:** Trigger set $\tilde{D} = \{(\tilde{x}_i, \tilde{y}_i)\}$;

**1 for all** $j = 1, \cdots, s$ **do**

**2**      Generate perturbed model $M$ ;

**3**      **if** $|para(M_b) - para(M)| \leq \delta$ **then**

**4**          $M_{b_j} = M$ ;

**5 for all** $i = 1, \cdots, m$ **do**

**6**      Choose $\tilde{x}_i$ such that top-1 probability of $M_b(\tilde{x}_i) \geq a$ and top-2 probability of $M_b(\tilde{x}_i) \geq b$ ;

**7**      $\tilde{y}_{i_j} = M_{b_j}(\tilde{x}_i)$ for all $j \in [s]$ ;

**8**      $\tilde{y}_i = \arg\min_{y \in \{1, \cdots, K\}} \left( \sum_{j=1}^{s} \mathbb{1}[\tilde{y}_{i_j} = y] \right)$ ;

**9**      $\tilde{D} = \tilde{D} \cup \{(\tilde{x}_i, \tilde{y}_i)\}$ ;

**10 return** Trigger set $\tilde{D} = \{(\tilde{x}_i, \tilde{y}_i)\}$;

---

of input samples for the trigger set. We initiate this process by training a base model, and select samples located near the decision boundary generated by the base model to construct the trigger set. This selection strategy is motivated by two key considerations: (1) When a sample is selected as a trigger, it tends to exert influence on neighboring samples, thus preventing its dismissal as an outlier. Notably, a substantial portion of the data clusters around the center, while samples near the boundary are sparser. By utilizing these samples as triggers, we minimize their impact on overall model accuracy. (2) Samples near the decision boundary exhibit labels that are more sensitive and prone to change. Consequently, alterations to trigger set labels can be readily accepted by stolen models, providing us with a strong criterion for identifying whether a model has been stolen.

Subsequently, we assign labels to the samples selected in the previous step. The labels we choose are designed to sufficiently identify benign and stolen models, aiming for a high probability of distinction, i.e., $\Pr[\text{diff}(\tilde{x}, \tilde{y})] \approx 1$. As illustrated in Algorithm 1, our method first perturbs the parameters of the base model $M_b$ within a certain range to create multiple variations $M_{b_1}, \cdots, M_{b_s}$. Then, we query these perturbed models with the selected sample $\tilde{x}_i$ to obtain $\tilde{y}_{i_1}, \cdots, \tilde{y}_{i_s}$, and choose the label $\tilde{y}_i$ that occurs least frequently among all predicted labels. This process can be expressed as

$$\tilde{y}_i = \underset{y \in \{1, \cdots, K\}}{\arg\min} \left( \sum_{j=1}^{s} \mathbb{1} \left[ M_{b_j}(\tilde{x}_i) = y \right] \right).$$

### 3.3 WATERMARK CERTIFICATION

In this section, we focus on certifying our proposed watermarking method. Rather than directly constraining the model parameters as in random smoothing methods, our proof emphasizes the activation of neurons associated with the watermark, allowing us to focus solely on watermark-related neurons.

First, recall that the trigger set is defined as $\tilde{\mathcal{D}} = \{(\tilde{x}_i, \tilde{y}_i)\}_{i=1}^{m}$. We define the event of successfully identifying the $i$-th watermark $(\tilde{x}_i, \tilde{y}_i)$ as $W_i$. Consequently, $\Pr[W_i]$ indicates the probability that the watermark $(\tilde{x}_i, \tilde{y}_i)$ is detected in the suspicious model.

When a watermark is successfully detected, a specific subset of neurons of the network is activated. We define the set of neurons activated by $(\tilde{x}_i, \tilde{y}_i)$ as $\Gamma(W_i) = \{W_{i,1}, \cdots, W_{i,s}\}$, where $W_{i,j}$ represents the $j$-th neuron activated in response to the $i$-th watermark. Additionally, $\Pr\left[\overline{W_{i,j}}\right]$ represents the probability that this neuron remains inactive.

Our method guarantees that if the neurons related to each watermark are not excessively likely to remain inactive (Equation 5), the probability of successfully detecting the watermark will be positive (Equation 6). Note that our proof focuses solely on the neurons associated with the watermark,

enabling us to establish a tight bound by imposing more precise requirements on these parameters. The main result is presented in the following theorem and see Appendix A for the complete proof.

**Theorem 2.** *Let $\mathcal{N}$ be the set of all neurons related to the watermarks, i.e., $\mathcal{N} = \bigcup_{i=1}^{m} \Gamma(W_i)$. If there exists an assignment of reals $\alpha : \mathcal{N} \to [0, 1)$ such that*

$$\Pr\left[\overline{W_{i,j}}\right] \leq \alpha\left(W_{i,j}\right) \prod_{W_{i,j'} \in \Gamma(W_i) \setminus \{W_{i,j}\}} \left(1 - \alpha\left(W_{i,j'}\right)\right), \quad \forall W_{i,j} \in \mathcal{N}, \tag{5}$$

*then the probability of avoiding all neurons remaining inactive is positive, in particular*

$$\Pr\left[W_i\right] = \Pr\left[\bigwedge_{W_{i,j} \in \Gamma(W_i)} W_{i,j}\right] \geq \prod_{W_{i,j} \in \Gamma(W_i)} \left(1 - \alpha\left(W_{i,j}\right)\right), \quad \forall i \in [m]. \tag{6}$$

Furthermore, given that our method selects labels with the highest distinguishability, we can derive insights into the size of the trigger set required to differentiate between benign and stolen models:

**Corollary 2.1.** *If the size of trigger set $m$ satisfy*

$$m \geq \prod_{W_{i,j}} \left(1 - \alpha\left(W_{i,j}\right)\right)^{-1}, \tag{7}$$

*then*

$$\Pr\left[\bigvee_{i=1}^{m} W_i\right] \geq 1$$

*and we can ensure with high probability that our watermark is guaranteed to be unremovable.*

Due to the high distinguishability of our method, this establishes the minimum size requirement for the trigger set. In our experiments, we use a generalized setting with $m = 100$ in our experiments. For more details and proof in Appendix A.

For the parameter $\alpha$, it represents the confidence in the successful transfer of the watermark, which is determined by balancing the watermark embedding method and the adversarial attack intensity. When the accuracy of the trigger set is high and the attack intensity is not strong, the corresponding $\alpha$ is small; therefore, the probability that the neurons remain inactive is also small. For a general approach, the probability of neurons remaining inactive can be approximated through local simulations of attacks, thereby assessing the probability of successful watermark transfer.

## 4 EXPERIMENTS

In this section, we evaluate our boundary-based watermarking method in comparison to several other baseline approaches, demonstrating its robustness against model extraction attacks.

### 4.1 EXPERIMENTAL SETUP

**Datasets.** We use the CIFAR-10 and CIFAR-100 (Krizhevsky et al., 2009) datasets. Additionally, our evaluation is broadened to ImageNet (Deng et al., 2009), a large-scale image dataset that challenges existing watermarking techniques. For CIFAR-10 and CIFAR-100, we split the training set into a train set and a validation set.

**Model Extraction Attack.** Following prior work (Jia et al., 2021), we train a surrogate model $\hat{M}$ on the surrogate dataset $\hat{D}$ under the following three settings:

- **Soft-label.** In this setting, we train the stolen model $\hat{M}$ as in Equation 1 where $M(x)$ and $\hat{M}(x)$ are $K$-dimensional vectors of class probabilities.
- **Hard-label.** Rather than using the $K$-dimensional vectors of class probabilities, we train the stolen model $\hat{M}$ as in Equation 1, where $M(x)$ and $\hat{M}(x)$ are one-hot vectors, having a value of 1 at the predicted label and 0 elsewhere.

| Method | Metric | Source model | Surrogate models | | |
| --- | --- | --- | --- | --- | --- |
| | | | Soft-label | Hard-label | RGT |
| DI (Maini et al., 2021) | | **92.03 ± 0.25** | 92.50 ± 0.17 | 92.27 ± 0.38 | 92.23 ± 0.59 |
| RS (Bansal et al., 2022) | CIFAR-10 | 84.17 ± 1.01 | 88.93 ± 1.18 | 89.62 ± 0.97 | 90.14 ± 0.08 |
| MB (Kim et al., 2023) | acc. (%) | 87.81 ± 0.76 | 91.17 ± 0.76 | 91.88 ± 0.40 | 93.05 ± 0.20 |
| MAT (Li et al., 2024) | | 86.10 ± 0.54 | 88.50 ± 1.02 | 85.40 ± 0.50 | 93.88 ± 0.35 |
| **Boundary-based (Ours)** | | 87.86 ± 2.06 | 90.64 ± 0.44 | 90.63 ± 0.32 | 92.42 ± 0.52 |
| DI (Maini et al., 2021) | | $10^{-3}$ | $10^{-2}$ | $10^{-2}$ | $10^{-2}$ |
| MB (Kim et al., 2023) | p-value | $10^{-12}$ | $10^{-8}$ | $10^{-8}$ | $10^{-8}$ |
| **Boundary-based (Ours)** | | $\mathbf{10^{-15}}$ | $\mathbf{10^{-11}}$ | $\mathbf{10^{-9}}$ | $\mathbf{10^{-8}}$ |
| RS (Bansal et al., 2022) | | 95.67 ± 4.93 | 7.67 ± 4.04 | 6.33 ± 1.15 | 3.00 ± 0.00 |
| MB (Kim et al., 2023) | Trigger set | 100.00 ± 0.00 | 82.00 ± 1.00 | 51.33 ± 4.93 | 72.67 ± 6.66 |
| MAT (Li et al., 2024) | acc. (%) | 96.88 ± 5.32 | 72.01 ± 3.53 | 50.13 ± 6.48 | 52.30 ± 3.05 |
| **Boundary-based (Ours)** | | **100.00 ± 0.00** | **86.50 ± 1.81** | **54.30 ± 5.90** | **73.80 ± 8.58** |

| Method | Metric | Source model | Surrogate models | | |
| --- | --- | --- | --- | --- | --- |
| | | | Soft-label | Hard-label | RGT |
| DI (Maini et al., 2021) | | **70.97 ± 0.74** | 72.70 ± 0.26 | 71.33 ± 0.31 | 72.87 ± 0.59 |
| RS (Bansal et al., 2022) | CIFAR-100 | 59.87 ± 2.78 | 65.66 ± 1.53 | 65.79 ± 0.39 | 64.99 ± 0.30 |
| MB (Kim et al., 2023) | acc. (%) | 62.13 ± 4.36 | 67.66 ± 0.36 | 70.65 ± 0.49 | 70.24 ± 0.46 |
| MAT (Li et al., 2024) | | 62.11 ± 1.67 | 59.00 ± 1.27 | 66.78 ± 1.00 | 72.73 ± 1.40 |
| **Boundary-based (Ours)** | | 65.90 ± 6.52 | 68.52 ± 2.03 | 69.20 ± 0.55 | 71.85 ± 0.84 |
| DI (Maini et al., 2021) | | $10^{-3}$ | $10^{-2}$ | $10^{-2}$ | $10^{-2}$ |
| MB (Kim et al., 2023) | p-value | $10^{-10}$ | $10^{-7}$ | $10^{-6}$ | $10^{-6}$ |
| **Boundary-based (Ours)** | | $\mathbf{10^{-12}}$ | $\mathbf{10^{-10}}$ | $\mathbf{10^{-7}}$ | $\mathbf{10^{-6}}$ |
| RS (Bansal et al., 2022) | | 99.00 ± 1.73 | 2.67 ± 1.53 | 4.33 ± 4.16 | 2.00 ± 1.00 |
| MB (Kim et al., 2023) | Trigger set | 100.00 ± 0.00 | 70.67 ± 7.57 | 40.00 ± 8.89 | 62.66 ± 10.12 |
| MAT (Li et al., 2024) | acc. (%) | 68.14 ± 10.16 | 72.98 ± 11.34 | 29.43 ± 6.58 | 35.73 ± 9.43 |
| **Boundary-based (Ours)** | | **100.00 ± 0.00** | **75.80 ± 5.32** | **41.32 ± 2.68** | **62.68 ± 3.54** |

Table 1: Results for watermarking DNNs against model extraction attacks on CIFAR-10 and CIFAR-100, where the scores for the best performance are bolded.

- **Regularization with Ground Truth Label.** We train the stolen model by simultaneously minimizing the empirical loss on the training dataset $D_c$ and the KL-divergence between the outputs of the source model and the stolen model:

$$\mathcal{L}_{RGT}(D_c, \hat{D}) = \beta \mathcal{L}(D_c) + (1 - \beta)\hat{\mathcal{L}}(\hat{\mathcal{D}}) \tag{8}$$

where $\beta \in [0, 1]$ is a hyperparameter to control the adversary's preference. Since the adversary has access to the ground truth labels, this attack method makes it easier to remove the watermark embedded in the source model.

**Baselines.** We compare our method against the following baselines.

- **Dataset Inference (DI)** (Maini et al., 2021). This method calculates a proxy margin between each sample and the decision boundaries of each class to create a margin embedding. The embedding is then used to train a binary classifier, and a t-test is finally performed on the classifier's confidence scores to verify the authenticity of the suspicious model.
- **Multi-View Data (MAT)** (Li et al., 2024). This paper presents an approach to trigger set-based watermarking by leveraging multi-view data. MAT embeds watermarks in DNNs by constructing a multi-view trigger set and using feature-based regularization during training.
- **Randomized Smoothing (RS)** (Bansal et al., 2022). It applies randomized smoothing to the parameters of the source model, ensuring that watermarks cannot be removed through small modifications to the model's parameters.
- **Margin-based Watermarking (MB)** (Kim et al., 2023). It uses projected gradient descent to maximize the margin of samples in the trigger set.

**Metric.** We measure the accuracy of the surrogate model on the trigger set $\tilde{\mathcal{D}}$ to evaluate the performance of the watermarking. Additionally, we compute the p-value from the t-test for statistical

| Method | Source Model ImageNet acc. (%) | Soft-label Trigger Set acc. (%) | p-value | Hard-label Trigger Set acc. (%) | p-value |
|---|---|---|---|---|---|
| MB (Kim et al., 2023) | $64.00 \pm 5.76$ | $13.86 \pm 0.56$ | $10^{-2}$ | $5.76 \pm 0.31$ | $10^{-1}$ |
| **Boundary-based (Ours)** | **$65.90 \pm 13.31$** | **$18.16 \pm 1.09$** | **$10^{-3}$** | **$13.77 \pm 0.86$** | **$10^{-3}$** |

Table 2: Results for watermarking DNNs against model extraction attacks on ImageNet, where the scores for the best performance are bolded.

testing methods, where a small p-value indicates distinguishability between the stolen and benign models.

**Settings.** We train base models using ResNet34 (He et al., 2016), achieving accuracies of 94.9% on CIFAR-10 and 75% on CIFAR-100. Based on these models, we select $m = 100$ samples $\tilde{x}_i$ where the highest and second-highest predicted probabilities were close. For simulating $s = 100$ independently trained models, we perturb the parameters with the $\ell_2$ distance not exceeding 0.4 (Bansal et al., 2022), resulting in worst-case accuracies of 82.7% and 59.1% for CIFAR-10 and CIFAR-100, respectively. For selecting boundary samples $\tilde{x}$, we gradually adjust the top two probabilities in Algorithm 1 until we precisely identify 100 samples. Finally, we select the labels $\tilde{y}_i$ with the lowest occurrence probabilities to form the trigger set $\tilde{\mathcal{D}} = \{(\tilde{x}_i, \tilde{y}_i)\}_{i=1}^{m}$. We set the batch size of the trigger set to 25 and introduce perturbations during training to prevent the watermark from being removed as an outlier. ResNet34 is used for all the source and surrogate models, with each model trained for 200 epochs, and $\beta$ is set to 0.3 in Equation 8.

### 4.2 RESULTS ON FUNCTIONALITY STEALING

Table 1 shows quantitative comparison with state-of-the-art methods on CIFAR-10 and CIFAR-100 datasets. Our method achieves 100% trigger set accuracy in the source models, similar to the MB method (Kim et al., 2023). Additionally, our source models achieve 87.86% and 65.90% accuracy on CIFAR-10 and CIFAR-100 datasets, which demonstrates the effectiveness of the proposed selection strategy.

As presented in Table 1, our method achieves an optimal balance between accuracy on CIFAR-10 / CIFAR-100 and the trigger set, demonstrating that our approach enhances the discriminability of the trigger set while maintaining model performance. Surrogate models trained with soft-label and hard-label also exhibit significant improvements compared with the baseline methods. Moreover, surrogate models trained with RGT exhibit performance similar to prior state-of-the-art techniques, given the pivotal role of ground truth labels for the adversary in the context of trigger set evaluation. Similarly, as shown in Table 2, our method achieves increased trigger set accuracy and a reduced p-value on ImageNet, showcasing its scalability on large-scale datasets.

Our method also exhibits superior results in statistical testing experiments. It consistently produces the smallest p-values across all attacks, indicating the effectiveness of our approach in distinguishing benign and stolen models. Notably, each trigger sample in our design is based on the least probable labels, enabling the identification of the model ownership with 95% confidence even if the suspect model reveals only a few instances or a single watermark pattern.

### 4.3 HETEROGENEOUS SURROGATE DATASET & ARCHITECTURE

In this section, we explore a realistic scenario where the adversaries are unseen from either the training dataset $\mathcal{D}_c$ or the network architecture of the source model. We use the SVHN dataset (Netzer et al., 2011) as the surrogate dataset and maintain consistent settings to train a surrogate model through a model extraction attack employing the soft-label technique. For the model architecture, we employ VGG11 (Simonyan & Zisserman, 2014) in the surrogate model and perform a soft-label attack as the watermarked model trained on CIFAR-10.

Table 3 shows that our method outperforms the margin-based approach (Kim et al., 2023) in the trigger set accuracy in both scenarios. Nonetheless, when comparing the trigger set accuracy with that in Table 1, the accuracy decreases with the utilization of the new dataset and architecture, indicating that it is challenging for the surrogate model to generalize to the trigger set. Hence,

| Method | Source model | | Surrogate model |
| | Clean acc. (%) | Trigger set acc. (%) | Trigger set acc. (%) |
|---|---|---|---|
| | Surrogate Dataset with SVHN | | |
| MB (Kim et al., 2023) | $87.81 \pm 0.76$ | $100.0 \pm 0.00$ | $72.00 \pm 6.08$ |
| Boundary-based (Ours) | $88.62 \pm 2.10$ | $100.0 \pm 0.00$ | $77.03 \pm 5.32$ |
| | Surrogate Model with VGG11 | | |
| MB (Kim et al., 2023) | $87.81 \pm 0.76$ | $100.0 \pm 0.00$ | $32.00 \pm 7.21$ |
| Boundary-based (Ours) | $89.63 \pm 2.05$ | $96.05 \pm 0.35$ | $35.74 \pm 5.25$ |

Table 3: Results for watermarking DNNs against model extraction attacks with different surrogate datasets and architectures.

even if the accuracy does not reach its optimum, we can assert model ownership with over 95% confidence.

## 4.4 ABLATION STUDIES

In this section, we provide ablation experiments on the effectiveness of our trigger set selection strategy compared with alternative methods including center-oriented selection and random labeling strategies.

| Strategy | Source Model | | Surrogate Model |
| | Clean | Trigger Acc. (%) | Trigger Acc. (%) |
|---|---|---|---|
| Center | 86.80 | 95.60 | 75.00 |
| Ours | 87.86 | 100.00 | 86.50 |

Table 4: Ablation study of different trigger set selection strategies on the CIFAR-10 dataset.

| Strategy | Source Model | | Surrogate Model |
| | Clean | Trigger Acc. (%) | p-value |
|---|---|---|---|
| Random | 89.14 | 100.00 | $10^{-9}$ |
| Ours | 87.86 | 100.00 | $10^{-11}$ |

Table 5: Ablation study of different trigger set labeling strategies on the CIFAR-10 dataset.

First, we compare our sample selection approach with the center-oriented strategy that focuses on samples clustering around central points. As illustrated in Table 4, when compared with the center-oriented strategy, our approach demonstrates superior performance in accuracy of both the clean data and the trigger set.

Next, we delve into the commonly used strategy of assigning random labels to boundary samples. As shown in Table 5, this approach results in ambiguous labels, making it more challenging to distinguish benign and stolen models.

Finally, we train 50 base models using various seeds to evaluate the validity of the experimental setup for generating perturbed models. The experimental results reveal that when employing randomly assigned labels, the average accuracy reaches 7%, whereas our approach achieves only 1%. In this context, lower accuracy signifies higher distinguishability, verifying that our watermark rarely appears in benign models, rendering it highly effective at identifying benign and stolen models.

## 5 CONCLUSION

In this paper, we present a boundary-based watermarking method designed to defend against model extraction attacks by carefully selecting boundary samples and assigning permissive labels. Our approach dissects the identification of stolen models into two key components: trigger set discriminability and the probability of distinguishing benign and stolen models. Accordingly, we propose a trigger sample selection strategy to enhance both aspects. Furthermore, we establish a tight theoretical bound on watermarking models by constraining parameters using the Lovász Local Lemma, ensuring reliable watermark detection in stolen models through the incorporation of multiple pairs of watermarks. Extensive experiments on three public datasets demonstrate the superiority of our method on ownership verification of stolen models.

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

# A    PROOF OF CERTIFICATION WATERMARK

(Theorem 2). *Let $\mathcal{N}$ be the set of all neurons related to the watermarks, i.e., $\mathcal{N} = \bigcup_{i=1}^{m} \Gamma(W_i)$. If there exists an assignment of reals $\alpha : \mathcal{N} \to [0, 1)$ such that*

$$\Pr\left[\overline{W_{i,j}}\right] \leq \alpha\left(W_{i,j}\right) \prod_{W_{i,j'} \in \Gamma(W_i) \setminus \{W_{i,j}\}} \left(1 - \alpha\left(W_{i,j'}\right)\right), \quad \forall W_{i,j} \in \mathcal{N},$$

*then the probability of avoiding all neurons remaining inactive is positive, in particular*

$$\Pr\left[W_i\right] = \Pr\left[\bigwedge_{W_{i,j} \in \Gamma(W_i)} W_{i,j}\right] \geq \prod_{W_{i,j} \in \Gamma(W_i)} \left(1 - \alpha\left(W_{i,j}\right)\right), \quad \forall i \in [m].$$

*Proof.* First, let $\Gamma(W_i) = t$ and by chain rule, we have

$$\Pr\left[W_i\right] = \Pr\left[\bigwedge_{j=1}^{t} W_{i,j}\right]$$

$$= \prod_{j=1}^{t} \Pr\left[W_{i,j} \middle| \bigwedge_{k<j} W_{i,k}\right]$$

$$= \prod_{j=1}^{t} \left(1 - \Pr\left[\overline{W_{i,j}} \middle| \bigwedge_{k<j} W_{i,k}\right]\right).$$

We use the Induction Hypothesis (I.H.) for the proof. And we perform induction on the number of conditions $\ell$: suppose that for any distinct events $W_{i,j}, W_{i,j_1}, \cdots, W_{i,j_\ell}$, the following hypothesis holds:

$$\Pr\left[\overline{W_{i,j}} \middle| W_{i,j_1} \cdots W_{i,j_\ell}\right] \leq \alpha(W_{i,j}). \tag{9}$$

For the base case when $\ell = 1$, we have

$$\Pr\left[\overline{W_{i,j}}\right] \leq \alpha\left(W_{i,j}\right) \prod_{W_{i,j'} \in \Gamma(W_i) \setminus \{W_{i,j}\}} \left(1 - \alpha\left(W_{i,j'}\right)\right)$$

$$\leq \alpha\left(W_{i,j}\right)$$

where $\alpha(\cdot) \in [0, 1)$.

Now, assume as the induction hypothesis that for $\ell - 1$, the result holds. We will show that under this assumption, the result also holds for a trigger set of size $\ell$.

First, we divide $W_{i,j_1}, \cdots, W_{i,j_\ell}$ into two parts. Say $W_{i,j_1}, \cdots, W_{i,j_\nu} \in \Gamma(W_{i,j})$ and $W_{i,j_{\nu+1}}, \cdots, W_{i,j_\ell} \notin \Gamma(W_i)$, i.e. $W_{i,j_1}, \cdots, W_{i,j_\nu}$ are the neurons activated in response to the $i$-th watermark, while the remaining $W_{i,j_{\nu+1}}, \cdots, W_{i,j_\ell}$ are those that remain inactive.

For $\nu = 0$, the case is trivial:

$$\Pr\left[\overline{W_{i,j}} \middle| W_{i,j_1} \cdots W_{i,j_\ell}\right] = \Pr\left[\overline{W_{i,j}}\right]$$

$$\leq \alpha\left(W_{i,j}\right) \prod_{W_{i,j'} \in \Gamma(W_i) \setminus \{W_{i,j}\}} \left(1 - \alpha\left(W_{i,j'}\right)\right)$$

$$\leq \alpha\left(W_{i,j}\right)$$

where the first equality holds because $W_{i,j_1}, \cdots, W_{i,j_\ell} \notin \Gamma(W_i)$, which implies that $W_{i,j}$ is independent of $W_{i,j_1}, \cdots, W_{i,j_\ell}$.

For $\nu \geq 1$, we divide the conditions into two parts, neighbors and non-neighbors:

$$\Pr\left[\overline{W_{i,j}}\,\middle|\,W_{i,j_1}\cdots W_{i,j_\ell}\right] = \Pr\left[\overline{W_{i,j}}\,\middle|\,W_{i,j_1}\cdots W_{i,j_\nu}W_{i,j_{\nu+1}}\cdots W_{i,j_\ell}\right]$$

$$= \frac{\Pr\left[\overline{W_{i,j}}W_{i,j_1}\cdots W_{i,j_\nu}\,\middle|\,W_{i,j_{\nu+1}}\cdots W_{i,j_\ell}\right]}{\Pr\left[W_{i,j_1}\cdots W_{i,j_\nu}\,\middle|\,W_{i,j_{\nu+1}}\cdots W_{i,j_\ell}\right]}. \tag{10}$$

We compute the numerator and denominator in equation 10 separately. Let's first consider the numerator:

$$\Pr\left[\overline{W_{i,j}}W_{i,j_1}\cdots W_{i,j_\nu}\,\middle|\,W_{i,j_{\nu+1}}\cdots W_{i,j_\ell}\right] \leq \Pr\left[\overline{W_{i,j}}\,\middle|\,W_{i,j_{\nu+1}}\cdots W_{i,j_\ell}\right]$$

$$= \Pr\left[\overline{W_{i,j}}\right] \tag{11}$$

$$\leq \alpha\left(W_{i,j}\right) \prod_{W_{i,j'}\in\Gamma(W_i)\backslash\{W_{i,j}\}} \left(1-\alpha\left(W_{i,j'}\right)\right)$$

where the last inequality applies the condition 5 of the theorem.

Then, consider the denominator in equation 10:

$$\Pr\left[W_{i,j_1}\cdots W_{i,j_\nu}\,\middle|\,W_{i,j_{\nu+1}}\cdots W_{i,j_\ell}\right] = \prod_{r=1}^{\nu}\Pr\left[W_{i,j_r}\,\middle|\,W_{i,j_{\nu+1}}\cdots W_{i,j_\ell}\right]$$

$$= \prod_{r=1}^{\nu}\left(1-\Pr\left[\overline{W_{i,j_r}}\,\middle|\,W_{i,j_{\nu+1}}\cdots W_{i,j_\ell}\right]\right) \tag{12}$$

$$\geq \prod_{r=1}^{\nu}\left(1-\alpha(W_{i,j_r})\right)$$

$$\geq \prod_{W_{i,j'}\in\Gamma(W_i)\backslash\{W_{i,j}\}}\left(1-\alpha\left(W_{i,j'}\right)\right)$$

where the first inequality uses hypothesis 9, and the second inequality holds because $W_{i,j_1}\cdots W_{i,j_\nu} \in \Gamma(W_i)\backslash\{W_{i,j}\}$.

Substituting Equation 11 and 12 into 10,

$$\Pr\left[\overline{W_{i,j}}\,\middle|\,W_{i,j_1}\cdots W_{i,j_\ell}\right] = \frac{\Pr\left[\overline{W_{i,j}}W_{i,j_1}\cdots W_{i,j_\nu}\,\middle|\,W_{i,j_{\nu+1}}\cdots W_{i,j_\ell}\right]}{\Pr\left[W_{i,j_1}\cdots W_{i,j_\nu}\,\middle|\,W_{i,j_{\nu+1}}\cdots W_{i,j_\ell}\right]}$$

$$\leq \frac{\alpha\left(W_{i,j}\right)\prod_{W_{i,j'}\in\Gamma(W_i)\backslash\{W_{i,j}\}}\left(1-\alpha\left(W_{i,j'}\right)\right)}{\prod_{W_{i,j'}\in\Gamma(W_i)\backslash\{W_{i,j}\}}\left(1-\alpha\left(W_{i,j'}\right)\right)}$$

$$\leq \alpha(W_{i,j}).$$

With all assumptions confirmed, the hypothesis holds:

$$\Pr\left[\overline{W_{i,j}}\,\middle|\,W_{i,j_1}\cdots W_{i,j_\ell}\right] \leq \alpha(W_{i,j}).$$

Thus we have

$$\Pr\left[W_i\right] = \prod_{j=1}^{t}\left(1 - \Pr\left[\overline{W_{i,j}}\,\middle|\,\bigwedge_{k<j} W_{i,k}\right]\right)$$

$$\geq \prod_{W_{i,j}\in\Gamma(W_i)}\left(1 - \alpha\left(W_{i,j}\right)\right)$$

completing the proof. $\square$

(Corollary 2.1). *If the size of trigger set $m$ satisfy*

$$m \geq \prod_{W_{i,j}}\left(1 - \alpha\left(W_{i,j}\right)\right)^{-1},$$

*then*

$$\Pr\left[\bigvee_{i=1}^{m} W_i\right] \geq 1$$

*and we can ensure with high probability that our watermark is guaranteed to be unremovable.*

*Proof.* Since the total number of neurons $N$ is significantly greater than the number of watermark-related neurons, leading to $\mathcal{N} = O(\frac{1}{N})$. Thus we have

$$\Pr\left[\bigvee_{i=1}^{m} W_i\right] = \sum_{i=1}^{m}\Pr\left[W_i\right]$$

$$\geq m\cdot\min_{i}\left\{\prod_{W_{i,j}\in\Gamma(W_i)}\left(1 - \alpha\left(W_{i,j}\right)\right)\right\}$$

$$\geq \prod_{W_{i,j}}\left(1 - \alpha\left(W_{i,j}\right)\right)^{-1}\cdot\min_{i}\left\{\prod_{W_{i,j}\in\Gamma(W_i)}\left(1 - \alpha\left(W_{i,j}\right)\right)\right\}$$

$$\geq 1$$

Thus, we can ensures the watermark's reliability in detecting stolen models with a probability of $1 - O(\frac{1}{N})$, enabling model ownership identification with 95% confidence, even if the suspect model reveals only a few instances or a single watermark pattern. $\square$

