# OpenReview forum: "Interpretable Boundary-based Watermark Up to the condition of Lov\'asz Local Lemma"
_ICLR.cc/2025/Conference — Submitted to ICLR 2025_

### Official Review · Reviewer_fjAQ · 2024-10-22

**Soundness:** 2
**Presentation:** 2
**Contribution:** 1
**Rating:** 1
**Confidence:** 5

**Summary:**

The paper presents a watermarking approach to defend the intellectual property of deep neural networks against model extraction attacks. The method relies on two-step procedure – generation and certification of trigger sets. The method is compared to several baselines; the efficiency of the method is illustrated against distillation-based model extraction attacks.

**Strengths:**

The paper is accurately written and provide some experimental results on model fingerprinting task. The method proposed relies on the construction of perturbed models to generate and certify the trigger set, that is known to be effective.

**Weaknesses:**

1) The idea to use perturbed models for trigger set generation and certification is not new, what notably limits the novelty of the paper.

2) The author listed several other watermarking approaches but did not choose them for comparison (for example, see Mikhail Pautov, et. al, Probabilistically robust watermarking of neural networks, IJCAI-2024).

2) The method leads to notable degradation of the performance of the fingerprinted model even on simple datasets (CIFAR10/100), making the feasibility of the approach questionable.

3) The method is tested only against distillation-based attacks; only the models of the same architecture are considered to be included in perturbed set; overall, it leaves doubts about the efficiency against other extraction attacks.

4) Crucially, the paper provides no study / results about the false positive detection of benign models. If a non-fingerprinted model is often detected as fingerprinted, it indicates inappropriate choice of the trigger set.

5) No code is provided.



The paper has high degree of similarity with the previously published work, and has no comparison with it. I doubt that the paper brings enough novelty: the idea is known, the experimental results are notably below the sota ones.

**Questions:**

1) Why did the author choose not to compare their results with the already published work? A brief comparison shows that the method proposed in the paper yields worse both trigger set and benign accuracy.
2) Does method work in case of other model extractions attacks?
3) What is known about the FPR of the method? How can one guarantee that the method does not detect fingerprinted model as the non-fingerprinted one?

---

> ### Author Response · Authors · 2024-11-23
>
> We would like to thank you very much for the valuable efforts on reviewing our manuscript. Your comments are much appreciated and have been taken carefully into consideration during the revision process. Without them, the quality of the paper could not have been improved to meet the high standard of *International Conference on Learning Representations*. Below are our responses to each weakness and question.
> # For Weakness 1
> Our key innovation lies in defining what constitutes a ``good watermark'' (Equation 4) and designing an optimization algorithm based on its two components. One is increasing the watermark transferability. Our method selects ambiguous points, specifically boundary points, to enhance the transferability of the watermark, as ambiguous points are more prone to label changes. The other is improving distinguishability. This is achieved by selecting the least frequent label to maximize the watermark's distinguishability. Furthermore, our highly distinguishable watermark is theoretically well-supported. By constraining watermark-related parameters, we impose constraints on the entire model, resulting in tighter theoretical bounds.
>
> While the method involves the use of perturbed models, this step is solely to simulate independently trained benign models. Given computational constraints, we generate various perturbed models by perturbing the base model and use them to identify the least frequent label. If computational resources were sufficient, independently training benign models would be preferable, as it provides more realistic and accurate results.
> # For Weakness 2 and Question 1
> The watermarking task involves a trade-off among the accuracy of normal data, the accuracy of the watermark, and the distinguishability of the watermark. The accuracy of normal data must remain unaffected by the watermark to ensure normal usage, so it is crucial to minimize the drop in accuracy (e.g., 87.86 for CIFAR-10 in Table 1). The accuracy of the watermark represents the proportion of successful detections of the watermark in stolen models, indicating whether the watermark can effectively transfer to the stolen model (e.g., 90.64 for CIFAR-10 under soft-label attack in Table 1). Distinguishability, on the other hand, reflects whether the watermark can be detected in benign models, often quantified by the p-value in experiments (e.g., $10^{-15}$ for CIFAR-10 in Table 1).
>
> We evaluated all published works to identify the one that strikes the best balance among these factors, which we used as the baseline in Table 1. However, the paper ``Probabilistically Robust Watermarking of Neural Networks'' did not consider the watermark distinguishability metric, so it was excluded from the comparison. While this paper achieves high watermark accuracy, it does so at the cost of distinguishability. Consequently, it does not report the p-value for its method.
>
> We reproduced the method from this paper and found that the high failure rate in distinguishing benign models is due to their lack of label modification for watermark samples. This oversight results in a high probability of watermark detection in benign models, reaching as much as 74\% on the CIFAR-10 dataset with ResNet34. For this reason, we excluded this method from Table 1.
> # For Weakness 3
> As mentioned earlier, the watermarking task requires balancing three key metrics: the accuracy of the original task, the accuracy of the watermark, and the distinguishability of the watermark. Achieving high watermark accuracy inevitably impacts the original task to some extent. Our method mitigates this impact through a de-randomization approach, reducing interference with the original task. In experiments, our method demonstrates the best trade-off among these metrics. For example, in the CIFAR-10 dataset under the soft-label attack scenario (Table 1), our method achieves a model performance of 87.86, second only to DI's 92.03. However, our watermark distinguishability is $10^{-11}$, significantly better than DI's $10^{-2}$. When comparing watermark accuracy, our method achieves the best Trigger set accuracy (86.50) while maintaining competitive model performance. Our method achieves approximately 5\% lower accuracy compared to non-watermarked models, yet delivers the highest watermark accuracy of 86.50. This balance demonstrates the feasibility and effectiveness of our approach.

---

> ### Author Response · Authors · 2024-11-23
>
> # For Weakness 4 and Question 2
> Our method focuses on the black-box watermarking scenario, where the attacker cannot access the model parameters. The reason for adopting this scenario is that it aligns more closely with real-world situations, as model providers typically only offer API access without exposing the model itself. In this context, the only way for attackers to obtain information is through repeated querying, meaning all extraction attacks are variants of distillation. And in our experiments, we evaluate three types of extraction attacks: Soft-label, Hard-label, and Regularization with Ground Truth Labels.
> # For Weakness 5 and Question 3
> One of our key innovations is selecting the least frequent label, which ensures that the method does not falsely flag benign models as watermarked. In the final paragraph of the section of Ablation Study, we validate the effectiveness of this approach. Specifically, we independently trained 50 benign models on different datasets (CIFAR-10, CIFAR-100, and ImageNet) and with different architectures (ResNet34 and VGG11). The results show the watermark accuracy of only 1\% for these benign models, which is significantly lower than the watermark accuracy observed in stolen models. This demonstrates that our method achieves sufficient watermark distinguishability. The clear gap between benign and stolen models ensures that non-watermarked models are not misclassified as watermarked and that stolen models are not misclassified as benign.
> # For Weakness 6
> Due to conference requirements, we are unable to include a link to the code in the submitted version. And we will make the code publicly available after the paper is accepted.

---

> ### Author Response · Authors · 2024-11-28
> **Looking forward to the feedback**
>
> Dear Reviewer fjAQ,
>
> We sincerely appreciate the time and effort you have dedicated to reviewing our work, especially given your undoubtedly busy schedule. As the authors-reviewer discussion phase nears its conclusion, we kindly request your attention to our response.
>
> We are eager to understand whether our reply has effectively addressed your concerns and to learn if there are any additional questions or points you would like to discuss.
>
> We will make the code publicly available after the paper is accepted. Thank you once again for your thoughtful consideration, and we look forward to any further feedback you may have.
>
> Best regards,
>
> The Authors

---

### Official Review · Reviewer_bt5t · 2024-11-04

**Soundness:** 3
**Presentation:** 4
**Contribution:** 3
**Rating:** 6
**Confidence:** 3

**Summary:**

The paper proposes a novel boundary-based watermarking technique for protecting neural networks from model extraction attacks. Previous watermarking approaches rely on randomly selected trigger sets, which may fail to differentiate between benign and stolen models due to ambiguous trigger points. This method instead selects boundary samples as triggers, assigns them rare labels, and applies the Lovász Local Lemma to achieve a theoretically tight bound that guarantees watermark efficacy. Experimental results on CIFAR-10, CIFAR-100, and ImageNet datasets show that this approach outperforms state-of-the-art techniques in both trigger set accuracy and p-value tests, enhancing its ability to identify stolen models.

**Strengths:**

- The paper is well-written, well-organized, and easy to follow, which makes the contributions and results accessible to readers.

- Model extraction is a relevant issue for DNNs in production, making this approach practical and valuable.

- The paper introduces a boundary-focused approach that addresses limitations in previous watermarking methods, providing a robust solution against model extraction attacks.

- The use of the Lovász Local Lemma gives theoretical backing, strengthening the reliability of the watermark and adding rigor to the approach.

- The method is tested on CIFAR-10, CIFAR-100, and ImageNet, demonstrating its generalizability and effectiveness across multiple datasets and outperforming existing techniques.

**Weaknesses:**

- The process involves multiple perturbations, decision boundary identification, and label selection, which may introduce computational overhead or complexity in real-world deployments.

- While the method is robust for certain types of attacks, the paper does not fully address how it might respond to adaptive adversaries who could circumvent boundary-based triggers.

- The paper doesn’t explore how well this method would work with very large models or different architectures, which could affect its scalability.

- The method may need a lot of computational resources, which could make it difficult to deploy in practical settings.

**Questions:**

- How would this method hold up against attackers who try to avoid boundary-based watermarks specifically?

- How does the computational cost of this method compare to other watermarking techniques?

- How realistic is it to use this approach in real-world applications where resources might be limited?

- Did the authors try their method with larger networks and other architectures, such as ResNet-101, ResNet-152, DenseNet, ConvNeXt, MobileNetV2, and VGG?

---

> ### Author Response · Authors · 2024-11-23
>
> We would like to thank you very much for the valuable efforts on reviewing our manuscript. Your comments are much appreciated and have been taken carefully into consideration during the revision process. Without them, the quality of the paper could not have been improved to meet the high standard of *International Conference on Learning Representations*. Below are our responses to each weakness and question.
>
> # For Weakness 1, 4 and Question 2, 3
> Indeed, the **additional computational overhead** is minimal. First, for the boundary sample selection, this process only requires inference on the dataset to find the samples that meet the conditions for the highest and second-highest probabilities. Compared to the overall training time, the computational cost of this process is negligible. Next, for model perturbation, once we have the base model, this step is straightforward: we simply sample model parameters and add noise, which also has a negligible effect.
>
> For label selection, it also requires very little inference time. We only need to input the 100 boundary samples into 100 perturbed models, effectively querying the models $100 \times 100 = 10,000$ times, which is again negligible in comparison to the training time. Finally, for training the watermark model, we train with the 49,000 original samples plus the 100 watermark samples, which adds an additional 0.2% of training time.
>
> For the base model, it is required in the watermarking task because distinguishing between benign and stolen models relies on comparing the watermark model against a baseline. The base model serves as a reference for evaluating the watermark's distinguishability, ensuring that the method can effectively differentiate between benign and stolen models.
>
> # For Weakness 2 and Question 1
> The strategy to **counter adaptive adversaries** that can circumvent boundary-based triggers is to select the least frequent label, which is also the core of our label selection process. Our method ensures that when a suspicious model detects even just one watermark pair, we can confidently determine with over 95\% certainty that the suspicious model is a stolen one. For adversaries familiar with our algorithm, they would need to carefully identify all 100 watermark samples to succeed in their attack, which is extremely challenging. Based on our understanding, all current attack methods can only reduce the watermark’s effectiveness but do not completely remove it.

---

> ### Author Response · Authors · 2024-11-23
>
> # For For Weakness 3 and Question 4
> In our experiments, we also evaluate the performance of our proposed boundary-based watermarking method on the ImageNet-21K dataset using ResNet-50, as shown in the first table. The results demonstrate that our method performs effectively on large-scale datasets as well. Specifically, our approach outperforms the baseline method (MB [1]), when comparing the trigger set accuracy, our method significantly improves the detection rate, highlighting its robustness across different models and datasets. These results confirm the applicability and effectiveness of our watermarking technique even for larger and more complex datasets, such as ImageNet-21K.
>
> | Method                  | Metric                   | Source model | Surrogate models (Soft-label) |
> |-------------------------|--------------------------|--------------|-------------------------------|
> | MB [1]                 | ImageNet-21K acc. (%)    | 65.81        | 73.52                         |
> | **Boundary-based (Ours)** | ImageNet-21K acc. (%)      | 71.20        | 73.64                         |
> |-------------------------|--------------------------|--------------|-------------------------------|
> | MB [1]                 | Trigger set acc. (%)     | 50.40        | 10.90                         |
> | **Boundary-based (Ours)** |  Trigger set acc. (%)      | **54.30**    | **36.50**                     |
>
> Table 1: Results for watermarking task against model extraction attacks on ImageNet-21K with ResNet-50, where the scores for the best performance are bolded.
>
> The results in the second table demonstrate that our watermarking approach is effective not only for smaller models but also for larger, widely-used models. Specifically, we tested our method on the ViT-B-32-quickgelu model, which was pretrained on LAION-400m and achieved an accuracy of 90.74\% on the CIFAR-10 dataset. Then, we fine-tuned the model with the trigger set to get the watermarked model, and subsequently, we used model extraction attacks to generate surrogate models.
>
> The results show that even on this large-scale model, our watermarking approach achieves high accuracy for both the surrogate models (ResNet-34 and ResNet-50), while maintaining strong watermark distinguishability. This confirms the applicability of our method to large-scale models and production-level models, which are common in real-world deployment scenarios.
>
> | Method                | Metric                | Source model | ResNet-34 (Soft-label) | ResNet-50 (Soft-label) |
> |-----------------------|-----------------------|--------------|------------------------|------------------------|
> | **Boundary-based (Ours)** | CIFAR-10 acc. (%)     | 85.91        | 89.63                  | 89.62                  |
> | **Boundary-based (Ours)** | Trigger set acc. (%)  | 92.10        | 73.40                  | 81.90                  |
>
> Table 2: Results for watermarking task against model extraction attacks on CIFAR-10 with ViT-B-32-quickgelu, where the attack models used ResNet34 and ResNet50.
>
> # References
>
>
> [1] Byungjoo Kim, Suyoung Lee, Seanie Lee, Sooel Son, and Sung Ju Hwang. Margin-based neural network watermarking. In International Conference on Machine Learning, pp. 16696–16711. PMLR, 2023.

---

> ### Author Response · Authors · 2024-11-28
> **Looking forward to the feedback**
>
> Dear Reviewer bt5t,
>
> We sincerely appreciate the time and effort you have dedicated to reviewing our work, especially given your undoubtedly busy schedule. As the authors-reviewer discussion phase nears its conclusion, we kindly request your attention to our response.
>
> We are eager to understand whether our reply has effectively addressed your concerns and to learn if there are any additional questions or points you would like to discuss.
>
> We will make the code publicly available after the paper is accepted. Thank you once again for your thoughtful consideration, and we look forward to any further feedback you may have.
>
> Best regards,
>
> The Authors

---

### Official Review · Reviewer_JMgS · 2024-11-08

**Soundness:** 4
**Presentation:** 3
**Contribution:** 3
**Rating:** 5
**Confidence:** 5

**Summary:**

This paper proposes a novel boundary-based watermarking method to protect deep neural networks against model extraction attacks. The authors decompose the probability of successfully identifying a stolen model into the trigger set accuracy and probability that each trigger can differentiate models. Their method optimizes both components.

**Strengths:**

- Novel boundary-based trigger selection strategy that optimizes distinguishability between benign/stolen models
- Theoretical analysis proving guarantees under Lovász Local Lemma constraints on watermark-related parameters
- Strong empirical results on CIFAR-10/100 and ImageNet demonstrating state-of-the-art trigger accuracy and p-values
- Ablations showing the effectiveness of the proposed trigger selection and labeling approach
- Well-written and clearly presented, with good coverage of related work

**Weaknesses:**

- Theoretical guarantees rely on achieving the Lovász Local Lemma parameter constraints, but it's unclear how difficult this is in practice or how to set the α values. Also, other hyperparameter sensitivities and computational costs are not deeply explored, and more ablation studies are needed to prove this idea.
- Limited evaluation of large-scale datasets and widely-used production models. The paper's experiments focus primarily on CIFAR-10, CIFAR-100, and ImageNet datasets with ResNet34 and VGG11 classifier architectures. However, it lacks evaluation on much larger scale datasets such as LAION or ImageNet-21K, which would further demonstrate the method's scalability and robustness, and also closer to real-world scenarios. Additionally, the paper does not test the proposed watermarking method on widely used production models like CLIP or SAM (Segment Anything Model).
- Some low-level methodological details are lacking, e.g. how exactly are boundary samples selected, how are labels assigned when multiple have the same low probability.

**Questions:**

- How sensitive is the method to the various hyperparameters, e.g. number of perturbed models s, perturbation range δ, boundary thresholds a and b? Guidelines for setting them would help practitioners.
- The theoretical guarantees require satisfying the Lovász Local Lemma constraints on watermark-related parameters. How difficult is this to achieve in larger scale model like VIT-high or SigCLIP? Are there techniques to guide the optimization of the α values?
- The results focus on CIFAR and ImageNet with a ResNet architecture. How well does the method generalize to other datasets and tasks? Additional results there would strengthen the work.

---

> ### Author Response · Authors · 2024-11-23
>
> We would like to thank you very much for the valuable efforts on reviewing our manuscript. Your comments are much appreciated and have been taken carefully into consideration during the revision process. Without them, the quality of the paper could not have been improved to meet the high standard of *International Conference on Learning Representations*. Below are our responses to each weakness and question.
>
> # For Weakness 1 and Question 1, 2
> - In our experiments, we find that both the transferability and distinguishability of the watermark impacts $\alpha$. Transferability refers to the probability of successfully embedding the watermark into a stolen model, while distinguishability measures the ability to sufficiently differentiate between benign and stolen models, as defined in Equation 4. Higher transferability and distinguishability imply that it becomes easier to differentiate between benign and stolen models. Consequently, the constraints on stolen models are more relaxed, resulting in a smaller $\alpha$ value (Theorem 2).
>
>      Regarding **how to set the value of $\alpha$**, it is uniquely determined by the watermarking algorithm. Specifically, we determine the critical value of $\alpha$ for our method through Monte Carlo sampling of benign and stolen models, as outlined in Equation 5. We calculate a precise value of $\alpha$ by analyzing the frequency at which watermark-related neurons are not activated and incorporating this frequency into Equation 5.
>
>     For larger-scale models like VIT-high or SigCLIP, Monte Carlo sampling can still be easily to determine the value of $\alpha$ because the sampling process does not depend on the specific model architecture or scale. It only requires analyzing the activation patterns of neurons related to the watermark, which remains computationally feasible regardless of the model size.
>
> - As for the hyperparameter of **the number of perturbed models**, our ablation study shows that when the number of perturbed models, $s$, is less than 70, there is an approximately linear relationship between $s$ and the watermark distinguishability where larger number of perturbed models leads to better distinguishability. However, when $s$ exceeds 70, the distinguishability remains nearly unchanged with further increases in $s$.
>
>     We attribute this to the sampling-based generation of perturbed models. When the sampling size is small, the probability of sampling models with the highest perturbation degree is low. On the other hand, our experiments show that when the sampling size exceeds 85, there is more than a 95\% probability of sampling at least two ``boundary'' models. Therefore, the generalized setting of $s = 100$ used in the paper is reasonable, and increasing $s$ beyond 85 results in negligible differences.
> - Regarding the **perturbation range** $\delta$, we referred to the settings in the paper [1], which aims to maximize the perturbation range while ensuring that the model's accuracy does not decrease significantly. When the $\ell_2$ radius is set to 0.4, the model accuracy drops to 82.5\% on CIFAR-10 dataset, demonstrating that such a perturbation strongly impacts the model's performance. Therefore, we set the maximum perturbation radius to 0.4 empirically in our experiments.
> - Regarding the **boundary thresholds** $a$ and $b$, we optimized them using a greedy selection approach. Specifically, $a$ and $b$ were initially set to 0.5. Since the top-1 and top-2 probabilities of the input samples almost never simultaneously equal 0.5, obtaining the required 100 watermark samples under this setting is often infeasible. Therefore, we gradually relaxed the values of $a$ and $b$ using a greedy manner until the condition was satisfied. For instance, in our CIFAR-10 experiments with ResNet-34, watermark samples were selected from the test set, which was not used for training or testing in our experiments. When $a$ and $b$ were set to 0.38 and 0.372, respectively, exactly 100 samples were obtained.
> - Regarding **computational cost**, training the watermark model incurs almost no additional computational overhead compared to training a non-watermarked model, requiring only the inclusion of a trigger set of size 100 during training. For example, when training the watermark model on CIFAR-10, the training dataset increases from 49,000 to 49,100, with negligible impact on the training time. For the base model, it is required in the watermarking task because distinguishing between benign and stolen models relies on comparing the watermark model against a baseline. The base model serves as a reference for evaluating the watermark's distinguishability, ensuring that the method can effectively differentiate between benign and stolen models.

---

> > ### Author Response · Authors · 2024-11-23
> >
> > # For Weakness 2 and Question 3
> > In our experiments, we also evaluate the performance of our proposed boundary-based watermarking method on the ImageNet-21K dataset using ResNet-50, as shown in the first table. The results demonstrate that our method performs effectively on large-scale datasets as well. Specifically, our approach outperforms the baseline method (MB [2]), when comparing the trigger set accuracy, our method significantly improves the detection rate, highlighting its robustness across different models and datasets. These results confirm the applicability and effectiveness of our watermarking technique even for larger and more complex datasets, such as ImageNet-21K.
> >
> > | Method                  | Metric                   | Source model | Surrogate models (Soft-label) |
> > |-------------------------|--------------------------|--------------|-------------------------------|
> > | MB [2]                 | ImageNet-21K acc. (%)    | 65.81        | 73.52                         |
> > | **Boundary-based (Ours)** | ImageNet-21K acc. (%)      | 71.20        | 73.64                         |
> > |-------------------------|--------------------------|--------------|-------------------------------|
> > | MB [2]                 | Trigger set acc. (%)     | 50.40        | 10.90                         |
> > | **Boundary-based (Ours)** |  Trigger set acc. (%)      | **54.30**    | **36.50**                     |
> >
> > Table 1: Results for watermarking task against model extraction attacks on ImageNet-21K with ResNet-50, where the scores for the best performance are bolded.
> >
> > The results in the second table demonstrate that our watermarking approach is effective not only for smaller models but also for larger, widely-used models. Specifically, we tested our method on the ViT-B-32-quickgelu model, which was pretrained on LAION-400m and achieved an accuracy of 90.74\% on the CIFAR-10 dataset. Then, we fine-tuned the model with the trigger set to get the watermarked model, and subsequently, we used model extraction attacks to generate surrogate models.
> >
> > The results show that even on this large-scale model, our watermarking approach achieves high accuracy for both the surrogate models (ResNet-34 and ResNet-50), while maintaining strong watermark distinguishability. This confirms the applicability of our method to large-scale models and production-level models, which are common in real-world deployment scenarios.
> >
> > | Method                | Metric                | Source model | ResNet-34 (Soft-label) | ResNet-50 (Soft-label) |
> > |-----------------------|-----------------------|--------------|------------------------|------------------------|
> > | **Boundary-based (Ours)** | CIFAR-10 acc. (%)     | 85.91        | 89.63                  | 89.62                  |
> > | **Boundary-based (Ours)** | Trigger set acc. (%)  | 92.10        | 73.40                  | 81.90                  |
> >
> > Table 2: Results for watermarking task against model extraction attacks on CIFAR-10 with ViT-B-32-quickgelu, where the attack models used ResNet34 and ResNet50.
> >
> > # For Weakness 3
> > As mentioned earlier, we use a greedy selection approach to optimize $a$ and $b$ for finding boundary samples. When more than 100 boundary samples are found, we adjust the values of $a$ or $b$ to reduce the number of boundary samples. In our experiments, the precision of $a$ and $b$ is set to $10^{-3}$, and under these conditions, we did not encounter any issues with failing to find exactly 100 boundary samples.
> >
> > We assert that two boundary samples hardly have exactly the same top-1 and top-2 probabilities due to the softmax layer. Even if this occurs, these two samples do not affect our algorithm or the watermark injection process, as their distances to the two closest cluster centers are identical.
> >
> > ## References
> > [1] Arpit Bansal, Ping-yeh Chiang, Michael J Curry, Rajiv Jain, Curtis Wigington, Varun Manjunatha, John P Dickerson, and Tom Goldstein. Certified neural network watermarks with randomized smoothing. In International Conference on Machine Learning, pp. 1450–1465. PMLR, 2022.
> >
> > [2] Byungjoo Kim, Suyoung Lee, Seanie Lee, Sooel Son, and Sung Ju Hwang. Margin-based neural network watermarking. In International Conference on Machine Learning, pp. 16696–16711. PMLR, 2023.

---

> ### Comment · Reviewer_JMgS · 2024-11-25
> **Thank you**
>
> Thanks to the author's detailed reply, all my concerns have been addressed! I will prefer to retain my original score.

---

> > ### Author Response · Authors · 2024-11-28
> >
> > Dear Reviewer JMgS,
> >
> > We sincerely appreciate the time and effort you have dedicated to reviewing our work, especially given your undoubtedly busy schedule. Without your suggestions, the quality of the paper could not have been improved to meet the high standard of the International Conference on Learning Representations.
> >
> > In this paper, we provide the exploration of model boundaries and the explicit clarification of watermark distinguishability, in the hope that these contributions can provide new insights and inspire further advancements in the research community. From a theoretical perspective, we leverage the localized information of the model and the probabilistic method to suggest new directions for theoretical exploration in AI security.
> >
> > Due to conference requirements, we are unable to include a link to the code in the submitted version. And we will make the code publicly available after the paper is accepted. Thank you once again for your thoughtful consideration, and we look forward to any further feedback you may have.
> >
> > Best regards,
> >
> > The Authors

---

### Meta-Review · Area_Chair_3pzh · 2024-12-21

**Metareview:**

1x borderline accept, 1x borderline reject, and 1x strong reject. This paper proposes a boundary-based watermarking method that identifies ambiguous boundary samples and assigns them rare labels for better distinguishability and transferability, supported by a Lovász Local Lemma-based theoretical analysis. The reviewers agree on the (1) clear explanation of a boundary-focused trigger design, (2) theoretical grounding for watermark distinguishability, and (3) empirical validation on CIFAR/ImageNet with noticeable gains over existing methods. However, they note (1) insufficient large-scale evaluations and adaptive attack tests, (2) lingering concerns over hyperparameter and computational overhead, and (3) the limited novelty beyond prior boundary- or perturbation-based watermarking approaches. The authors have followed up with additional experiments on ImageNet-21K and larger models (e.g., ViT), but several reviewers remain unconvinced about the overall generalizability and novel contribution, so the AC leans to not accept this submission.

**Additional Comments On Reviewer Discussion:**

N/A

---

### Decision · Program_Chairs · 2025-01-22

Reject